# *Haplosporidium pinnae* Detection from the Faeces of *Pinna nobilis*: A Quick and Noninvasive Tool to Monitor the Presence of Pathogen in Early-Stage or during Fan Mussel Mass Mortalities

Chiara Manfrin [1,*], Saul Ciriaco [2,3], Marco Segarich [3], Andrea Aiello [1], Fiorella Florian [1], Massimo Avian [1], Antonio Terlizzi [1,4], Piero G. Giulianini [1], Maurizio Spoto [2] and Alberto Pallavicini [1,4]

1   Dipartimento di Scienze della Vita (DSV), Università di Trieste, Via L. Giorgieri 5, 34127 Trieste, Italy
2   WWF AMP Miramare, Via Beirut 2/4, 34151 Trieste, Italy
3   Shoreline Soc. Coop., AREA Science Park, Padriciano, 99, 34149 Trieste, Italy
4   Stazione Zoologica Anton Dohrn, Villa Comunale, 80121 Napoli, Italy
*   Correspondence: cmanfrin@units.it; Tel.: +39-040-558-8617

**Abstract:** Due to the increasing mass mortality of *Pinna nobilis*, mainly caused by the protozoan *Haplosporidium pinnae* along the Mediterranean Sea, it is necessary to develop rapid and effective methods to detect the pathogen. The present study describes the development and validation of a species-specific assay based on hydrolysis probe chemistry to detect *H. pinnae* DNA from faeces and pseudofaeces of *P. nobilis*. During a study campaign in the Gulf of Trieste (Italy) in the spring and summer of 2022, 18 samples (10 faeces and 8 pseudofaeces) were collected. DNA was isolated from all samples and the presence of *H. pinnae* was tested by amplifying a small portion of 18S rDNA using qPCR. The newly developed assay detected positive *H. pinnae* in the faeces of the fan mussel in the spring, while no evidence of an outbreak of *H. pinnae* was found in the summer. In addition, the method proved to be noninvasive and can be used to monitor suspected *H. pinnae* infections in the early stages when bivalves are still vital. Furthermore, fecal analysis allows the monitoring of *P. nobilis* without dissecting tissues. The presented assay can also be used to routinely monitor the progress of mass mortalities caused by *H. pinnae* and to screen for the pathogen in live fan mussels and other environmental matrices, such as water, sediment, and faeces from other species that can host the protozoan.

**Keywords:** *Pinna nobilis*; *Haplosporidium pinnae*; faeces; 18SSU rRNA; eDNA

## 1. Introduction

Pathogens play an important role in regulating host population dynamics and determining the structure of marine communities [1,2]. The current warming trend could lead to changes in the basic biological characteristics of many marine populations, making them more susceptible to disease or promoting the spread of alien species with potentially pathogenic functions [3]. *Pinna nobilis* (Linnaeus, 1758) is the largest endemic Mediterranean bivalve, usually inhabiting the infralittoral plain on sandy or rocky substrates [4,5]. The importance of this species is related to its role as habitat creator and ecosystem engineer, as a natural filter of seawater, and to the symbiotic relationships that develop between the bivalve and some decapod crustaceans; by retaining large amounts of organic matter from suspended sediments, it contributes to water clarity, and its surfaces are heavily colonized, increasing the number of benthic species and specific local richness. This species has long been threatened by marine pollution, which affects the development of its larval stages, and illegal collection [6]. Recently, populations of *P. nobilis* have declined sharply [5], leading to the inclusion of this species as one of the protected species in the EU Habitat Directive (92/43/EEC, Annex IV), in the Protocol on Protected Areas and Biological Diversity in

the Mediterranean of the Barcelona Convention (Annex II), and the national legislation of many Mediterranean countries [7].

Starting from 2016 an unusual mass mortality event (MME) with a mortality rate of nearly 100% has been recorded along the whole Spanish Mediterranean coast, and subsequently on other coasts of the Mediterranean basin, affecting specimens of *Pinna nobilis* of all sizes, depths ranges, and habitat types [8]. In 2018, Catanese and colleagues [9], following histological and molecular analyses, indicated a haplosporidan parasite, later named *Haplosporidium pinnae*, as the likely cause of the fan mussel mass mortality [10] which is supposed to be an alien species introduced at that time in the Mediterranean basin [7]. Subsequent studies have shown the presence of *Mycobacterium* spp. and *Vibrio* spp. in addition to the presence of *H. pinnae*, indicating them as other possible causes of *Pinna nobilis* mortality events [11,12]. In December 2019, *P. nobilis* status on the IUCN red list was raised to critically endangered.

The present study describes the development and validation of a species-specific assay based on hydrolysis probe chemistry aimed at detecting *H. pinnae* DNA from the faeces of *P. nobilis* to spread a noninvasive detection method that allows the monitoring of living fan mussels.

The use of noninvasive molecular techniques represents a major advance in environmental biology, especially in biodiversity conservation, through analyses conducted in traces of organisms rather than whole organisms [13]. Among these methodologies, analysis through environmental DNA (eDNA) appears to be of recent interest not only because of its noninvasiveness toward organisms but also because of the reduced costs and time required for sampling [14]. This technique allows the monitoring of the presence/absence of one or multiple target species without the need to capture the target organisms, but by analyzing the DNA present in environmental samples (e.g., water, sediments, or air, and in our case faeces) [15]. Being the eDNA present in trace, detection methods must be highly sensitive, and at the same time protocols must be put in place to reduce the risk of false-negative and false-positive results, including the establishment of limits of detection (LODs) [16] and enzymatic inhibition control [17–19].

The presence of *H. pinnae* in one out of 18 samples demonstrates the high potential of the assay in detecting pathogens even in nonobvious outbreaks in an area where an outbreak of *H. pinnae* was reported in 2019 and detected in the mantle tissue of dead specimens [20].

## 2. Materials and Methods

### 2.1. Samples Collection, DNA Extraction, and Amplification

The samples were obtained at three different sites in the Gulf of Trieste: Miramare (Marine Protected Area), Sistiana, and Panzano (see Figure 1). Faecal and pseudofaecal samples were collected from individual specimens (shell length from 30 to 40 cm) by divers of the Miramare Marine Protected Area (Trieste, Italy) using Luer-Lok™ disposable syringes of 50 mL (BDPlastipak™, Eysins, Switzerland) without a needle. To collect the faecal samples, the divers gently touched the shell of the mollusc, to make them close and eject the faeces, which were immediately collected with the syringe. Underwater, the samples were transferred from the syringe to a vial (5 mL) and delivered refrigerated to the Genetics and Zoology Laboratory of the Department of Life Sciences on the same day. Samples were centrifuged at $13,000 \times g$ for 5 min, the water was discarded, and the faecal samples were stored at $-20$ °C until DNA extraction. Each faecal sample was collected from a different specimen of *P. nobilis*. Table 1 shows the list and the locations of the harvested samples.

DNA from faeces and pseudofaeces was extracted with Mag-Bind® Environmental DNA kit (Omega bio-tek, Norcross, GA, USA) following the protocol for soil present in the manufacturer's guidelines.

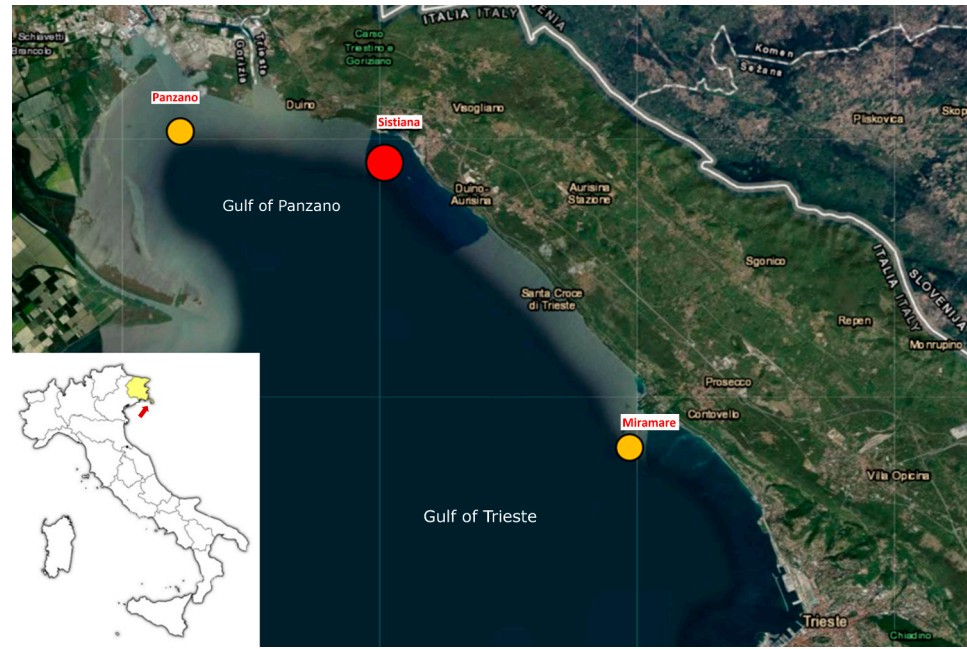

**Figure 1.** Map of Friuli Venezia Giulia region (a yellow region in figure Italy, the red arrow indicates monitored coast) showing stations where faecal and pseudofecal samples of *P. nobilis* were collected. In the samples collected in Panzano and Miramare, no *H. pinnae* DNA was detectable (sites indicated as yellow spot), while in Sistiana a positive result was obtained (shown as red spot).

**Table 1.** Details of samples collected in the Gulf of Trieste alongside the date, collection place, and geographic coordinates.

| ID Sample | Type | Collection Date | Place | Geographical Coordinates |
| --- | --- | --- | --- | --- |
| F1 | Faeces | 12 July 2022 | Miramare | 45.706739 N, 13.703184 E |
| F2 | Faeces | 12 July 2022 | Miramare | 45.706739 N, 13.703184 E |
| F3 | Faeces | 12 July 2022 | Miramare | 45.706739 N, 13.703184 E |
| F4 | Faeces | 22 August 2022 | Panzano | 45.781455 N, 13.548151 E |
| F5 | Faeces | 22 August 2022 | Panzano | 45.781455 N, 13.548151 E |
| F6 | Faeces | 22 August 2022 | Panzano | 45.781455 N, 13.548151 E |
| F7 | Faeces | 5 May 2022 | Sistiana | 45.762826 N, 13.632380 E |
| F8 | Faeces | 5 May 2022 | Sistiana | 45.762826 N, 13.632380 E |
| F9 | Faeces | 5 May 2022 | Sistiana | 45.762826 N, 13.632380 E |
| F10 | Faeces | 5 May 2022 | Sistiana | 45.762826 N, 13.632380 E |
| PS1 | Pseudofaeces | 12 July 2022 | Miramare | 45.706739 N, 13.703184 E |
| PS2 | Pseudofaeces | 12 July 2022 | Miramare | 45.706739 N, 13.703184 E |
| PS3 | Pseudofaeces | 5 May 2022 | Sistiana | 45.762826 N, 13.632380 E |
| PS4 | Pseudofaeces | 22 August 2022 | Panzano | 45.781455 N, 13.548151 E |
| PS5 | Pseudofaeces | 22 August 2022 | Panzano | 45.781455 N, 13.548151 E |
| PS6 | Pseudofaeces | 22 August 2022 | Panzano | 45.781455 N, 13.548151 E |
| PS7 | Pseudofaeces | 22 August 2022 | Panzano | 45.781455 N, 13.548151 E |
| PS8 | Pseudofaeces | 22 August 2022 | Panzano | 45.781455 N, 13.548151 E |

Amplification mix was set up as follows: 1X KAPA Probe Force master mix (Kapa Biosystems, Wilmington, MA, USA), 0.4 µM each primer, and 0.2 µM probe, 2 µL of eDNA,

and water at a final volume of 15 μL. Each field sample was run at a minimum of 3 technical replicates. The qPCR to evaluate the samples was set up with 5 NTC samples and 3 positive control samples consisting of genomic DNA from *P. nobilis* collected in the Gulf of Trieste in October 2019 and assessed positive to *H. pinnae* with the assay of 165 bp published by López-Sanmartín and colleagues [21] and with the here presented Hp_CM assay.

The thermal profile was as follows: 98 °C 3′, 50 cycles at 95 °C 10″ and 65 °C 30″ and qPCR was performed on a C1000 thermocycler equipped with a CFX96 head (Bio-Rad, Hercules, CA, USA). Quantitative analysis was run with the CFX Maestro software 1.1. v 4.1 (Bio-Rad, Hercules, CA, USA).

One replica of the positive sample was purified with 1.8X Mag-Bind® TotalPure NGS (Omega Bio-tek, Norcross, GA, USA) and sent to an external Sanger sequencing service (Eurofins Genomics, Ebersberg, Germany).

### 2.2. Primer Design

Searching at the NCBI website with the words: *"18S"* and *"Haplosporidium"*, we retrieved 9 sequences and with *"small subunit"* and *"Haplosporidium"*, we retrieved 122 sequences (up to 31 January 2023). All the sequences were aligned with MEGAX, v.10.1.1. software, and the alignment, alongside their GenBank IDs, is available as Supporting Information S1.

The specific probe and oligonucleotides were designed to amplify a fragment of 102 bp of the 18S rDNA of *H. pinnae* using Primer3Plus (http://www.bioinformatics.nl/cgi-bin/primer3plus/primer3plus.cgi, accessed on 30 January 2020) and OligoCalc (http://biotools.nubic.northwestern.edu/Oligo Calc.html, accessed on 30 January 2020) to exclude possible dimers and hairpin formations between the primers (Table 2).

**Table 2.** Oligonucleotides and probe sequences for *Haplosporisium pinnae* 18SSU rRNA fragment and their melting temperatures.

| Name | Sequence (5′->3′) | Temp Melting (°C) |
|---|---|---|
| Hp_CM_For | TGACTCAACACGGGGAAAC | 59.51 |
| HP-CM_probe | FAM-CCAGGGCCAGACATAGCCAGGA- BHQ1 | 69.74 |
| Hp_CM_Rev | ACTAAGAACGGCCATGCAC | 58.74 |

### 2.3. Limit of Detection and Inhibition Test

To estimate the limit of detection (LOD), the positive sample was serially diluted as follows: used as it is and dilutions from 1e-01 up to 1e-06 of the initial concentration. Each dilution was amplified in 5 replicates alongside 5 replicates of NTC. LOD was identified as the lowest concentration of the target analyte that can be detected with a 95% detection rate as the standard confidence level by using the R script for LOD calculation [16] in R Studio v. 2022.12.0+353.

For the samples collected in the field, the detection of *H. pinnae* eDNA was considered as "strong evidence" when at least two technical replicates were positive and showed a sigmoidal amplification curve (up to 45 Ct, [22]); whereas it was considered "suspected eDNA presence" when only one of the techniques showed at least one positive technical replicate, and "absent eDNA" when the sample was negative.

Furthermore, to evaluate possible inhibitor effects on the amplification of samples, an "internal amplification control" (IAC) was produced following the instruction in [17], namely IAC 1, alongside a HEX-probe designed in this study (Table 3). First, double-strand IAC (dsIAC) was generated in a 25 μL PCR mix containing 1X AccuStart II PCR ToughMix (Quantabio, Beverly, MA, USA), 0.4 μM each primer, and 50 ng of single-strand (ssIAC), by using the following thermal profile: initial denaturation 95 °C for 2′, 35 cycles: 95 °C for 30″, 61 °C for 30″, 72 °C for 1′, and a final extension of 72 °C for 10′. The product was checked in a 3% TAE Meta-Phor agarose gel (Lonza, Siena, Italy) and purified with 1.7×

Mag-Bind® TotalPure NGS (Omega bio-tek, Norcross, GA, USA). The purified product was quantified in triplicate with a Qubit ® dsDNA BR Assay Kit (Thermofisher, Waltham, MA, USA). The dsIAC was run in triplicate alongside a standard dilution series from $3 \times 10^5$ to $3 \times 10$ DNA copy number. Each reaction was performed in a final volume of 15 μL containing 1X KAPA Probe Force master mix (Kapa Biosystems, Wilmington, MA, USA), 0.3 μM of both forward and reverse primers, and 0.15 μM of the probe, designed by us on the IAC 1 sequence (Table 3) and 1 μL selected IAC dilution, by using the following thermal profile: initial denaturation 95 °C for 2′, 40 cycles: 95 °C for 30″, and 65 °C for 30″. Triplicates of the PCR reactions containing $3 \times 10^2$ IAC copy numbers and 1 μL of positive eDNA sample alongside samples randomly taken from the sampled stations were run for inhibition tests.

**Table 3.** Oligonucleotides and probe sequences for IAC 1 fragment and their melting temperatures.

| Name | Sequence (5′->3′) | Temp Melting (°C) |
|---|---|---|
| IAC 1_For | CCTCTGCAGCGATGTCACTA | 60.16 |
| IAC 1_probe | HEX- CCCGTCTAACACACCTGTCCA- BHQ1 | 63.26 |
| IAC 1_Rev | GCATCTGTCGGTGCGTCAAT | 64.54 |

## 3. Results

### 3.1. Haplosporidium Pinnae Assay

After downloading a total of 131 sequences related to the 18S rRNA sequence of *Haplosporidium* sp., 100 sequences were aligned, whereas the remaining 31 sequences were discarded since they did not overlap with the target region of the Hp_ CM _assay.

The forward primer matched 100% with sequences from *H. pinnae* and one sequence from *H. lusitanicum* (AY449713). The reverse primer matched 100% with most (92%) sequences but not with *H. lusitanicum* (AY449713), *H. pickfordi* (AY452724), *H. tuxtlensis* (JN368430), *H. littoralis* (KJ150289, JX185413), *H. cranc* (MN846352, MT311214), and *H. louisiana* (U47851). The probe matched 100% only with *H. pinnae* sequences and never with other species.

### 3.2. Amplification of H. pinnae DNA from Faeces and Pseudofaeces

Only one sample out of 18 (F8) was amplified and revealed strong evidence of *H. pinnae* DNA (six positives out of six replicates, 100%) presence in an alive fan mussel located in the Sistiana bay (Trieste, Italy). Confirmation of the *Haplosporidium pinnae* in the faeces was acquired other than the fluorescence signal in qPCR, but also with the Sanger sequencing of one of the replicas obtained for F8 (Figure S3). No positive signal or "suspected eDNA presence" was detected in the other faeces and among pseudofaeces samples. No amplification was observed on a total of five NTCs, while all the replicates of gDNA, as a positive control sample, were amplified (Table 3).

### 3.3. Limit of Detection and Inhibition Test

The lowest standard, with 95% or greater detection, was the F8 dilution of $10^4$. Figure 2 shows the outputs generated in R with the script of Klymus and colleagues [16].

Concerning the inhibition test, the differences in Ct values between IAC 1 and IAC 1 with a positive sample (F8) or with a sample in which *H. pinnae* DNA was not detected (F6) were compared for the same IAC1 dilution and the results were 0.49 (dilution $3 \times 10$ IAC 1 copy number + F8), 0.93 ($3 \times 10^2$ + F8) and 0.64 Ct ($3 \times 10^2$ + F6), respectively (Table 3). The standard curve for the inhibition test is provided in S2, in Supplementary Information.

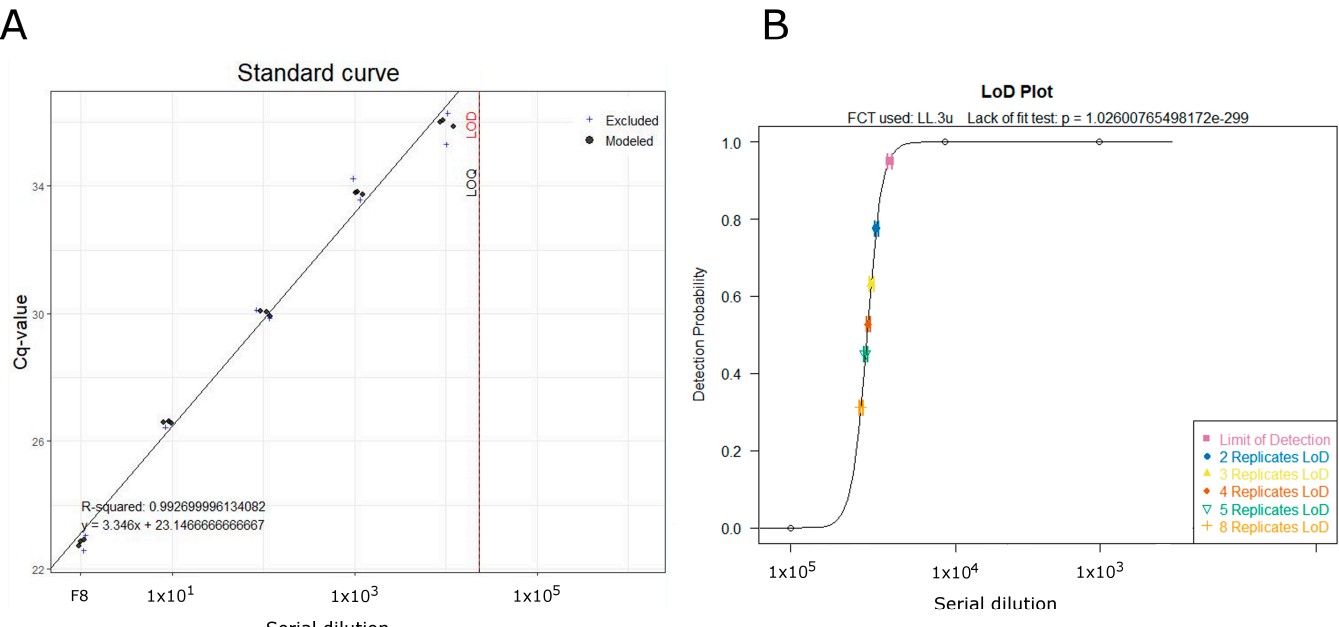

**Figure 2.** Output from Klymus script with minor modification. (**A**) Calibration curve plot with quantitative cycles (Cq) value on the y-axis and serial dilution of F8 sample (from as it is F8 up to a dilution of $1 \times 10^{-5}$) on the x-axis. Points drawn with black circles are the middle 2 quartiles of standards with $\geq 50\%$ detection and are included in the linear regression calculations. Points drawn with blue pluses (+) are outside the middle 2 quartiles or for standards with <50% detection and are not included in the linear regression. R-squared = coefficient of determination and y-int = y intercept. Additional dilutions of $10^5$ and $10^6$ are not visible in the plot since they were not amplified. (**B**) Limit of detection (LOD) plots. Detection probability is on the y-axis and serial dilutions of the F8 sample (from as it is F8 up to a dilution of $1 \times 10^{-5}$) are on the x-axis. Points are drawn with open circles for the detection rates of each standard tested, and the line represents the LOD model. Colored points with 95% confidence intervals are drawn to represent the LOD and effective LODs for multiple replicate analyses. The logarithmic function used and the lack of fit test results are shown in the subtitle.

## 4. Discussion

Due to the increasing spread of *H. pinnae* within the Mediterranean basin and the consequent mortality of *P. nobilis*, it is urgent to monitor the ongoing situation in real time. Assays and approaches to do that are increasingly available [21,23], and the present work shows the effectiveness of an eDNA approach to detect *H. pinnae* on the faeces.

The 100% match of all oligonucleotides belonging to the Hp_CM_assay occurred only in the *H. pinnae* sequences.

The positive signal was found in the F8 sample of an alive fan mussel in Sistiana Bay and was the only positive sample among the 10 faeces examined (see Cq in Table 4). Confirmation of the presence of *H. pinnae* was obtained with additional Sanger sequencing that confirmed the positive hit to *H. pinnae* (Figure S3). However, the pseudofaeces all gave a negative result in detecting *H. pinnae* DNA, which could be due to the nature of the pseudofaeces. Pseudofaeces are produced by filter-feeding bivalves (and filter-feeding gastropod molluscs) and are used to get rid of suspended matter that cannot be used as food and is rejected by the animals [24].

Since *H. pinnae* is a successful parasite of the fan mussel, it can likely escape initial control by the fan mussel during the injection and production of pseudofaeces. We hypothesize that if the concentration of *H. pinnae* in the water had been very high, the protozoan would likely have been detected in the pseudofaeces, though in our target study, we were likely not in a situation where the pathogen was present at high levels in the water.

**Table 4.** Average cycle thresholds (Ct) and standard deviation from the positive sample (F8), positive control sample, IAC 1 at two different dilutions, and IAC 1 at different dilution with F8 or F6 sample.

| Sample | Replicates | | | | | | | |
|---|---|---|---|---|---|---|---|---|
| | **Ct** | **Ct** | **Ct** | **Ct** | **Ct** | **Ct** | **Ct Mean** | **±SD** |
| F8 | 22.01 | 22.91 | 22.87 | 22.58 | 22.72 | 23.05 | 22.69 | 0.37 |
| Positive control | 25.41 | 22.05 | 23 | | | | 23.47 | 1.73 |
| IAC1 ($3 \times 10^2$) | 7.61 | 7.41 | 7 | | | | 7.34 | 0.31 |
| IAC1 ($3 \times 10$) + F8 | 7.85 | 7.81 | | | | | 7.83 | 0.03 |
| IAC1 ($3 \times 10^2$) | 11.31 | 10.94 | 10.87 | | | | 11.04 | 0.24 |
| IAC1 ($3 \times 10^2$) + F8 | 11.89 | 12.04 | | | | | 11.96 | 0.11 |
| IAC1 ($3 \times 10^2$) + F6 | 11.85 | 11.51 | | | | | 11.68 | 0.24 |

The LOD with a 95% detection rate for *H. pinnae* DNA was achieved at an F8 dilution of $10^4$, while further dilutions of $10^5$ and $10^6$ failed amplification. LOD was determined after the assay was optimized in terms of annealing temperature, primer/probe concentrations, etc., as these changes affect PCR efficiency and the limits of detection and quantification.

The positive detection of *H. pinnae* in the Bay of Sistiana (Trieste, Italy) is not surprising, as we already detected *H. pinnae* and *Mycobacterium* sp. in 18 dead but still bottom-adherent *P. nobilis* specimens in October 2019, which was the first *H. pinnae* outbreak in the northern Adriatic Sea ([20], document available upon request to authors). After this outbreak, only small patches of *P. nobilis* remained vital in the Gulf of Trieste. Therefore, the samples examined in this study were from the remaining live individuals that survived the *H. pinnae* infestation.

To date, there is only one other publication on the detection of *H. pinnae* from the faeces of *P. nobilis* [23], and this targets a fragment of 331 bp. Our Hp_CM assay amplifies a fragment of 102 bp and an amplicon size <150–200 bp, which should maximize the recovery of DNA from environmental samples [25]. Our assay can also be used on environmental matrices and on semidegraded material due to the shortness of the amplicon. Furthermore, the use of an eDNA approach is beneficial for routine monitoring because once the sample is collected, the result can be obtained in a short time interval (1–3 days) and the target is monitored without disturbing the *P. nobilis*, thus avoiding removing a small portion of its flesh and stressing a species that has already considerably declined.

The hope of finding *H. pinnae*-resistant populations is critical to the survival of this species. Further studies are needed to better define the genetic populations and the particular genetic traits that confer resistance to *H. pinnae* and to continue surveillance of this pathogen.

## 5. Conclusions

The results obtained support the use of the eDNA approach in the faeces of endangered species such as *Pinna nobilis*, where outbreaks of *H. pinnae* are currently observed since the approach is not invasive. Visual monitoring revealed that all specimens were alive during sampling, and the Hp_CM assay successfully detected *H. pinnae* during monitoring. This demonstrates the importance of continuous monitoring of live populations to predict the health status of the animals and to collect data on the possible presence of *H. pinnae*. Furthermore, the successful detection along with the size of the Hp_ CM assay lends itself to monitoring other environmental matrices such as water, soil, and faeces of other species living near *P. nobilis* populations or in environments where *H. pinnae* may be present to be used as sentinel organisms for detection of this protozoan. This approach is noninvasive and is also suitable for DNA amplification from semidecomposed material such as faeces or other environmental matrices.

**Supplementary Materials:** The following supporting information can be downloaded at: https://www.mdpi.com/article/10.3390/d15040477/s1. Figure S1: *Haplosporidium* sp. alignment with Hp_CM_assay. Figure S2: Standard curve plot on inhibition test, Figure S3: Alignment of the Sanger sequenced amplicon by F8 with other *Haplosporidium* spp. and along with Hp_CM oligonucleotides.

**Author Contributions:** Conceptualization. C.M., A.P, S.C. and M.S. (Maurizio Spoto); methodology. C.M., S.C., F.F. and M.S. (Marco Segarich); software. C.M.; validation. C.M. and A.P.; formal analysis. C.M. and A.A.; resources. A.P.; data curation. C.M.; writing-original draft preparation. C.M. and A.A.; writing-review and editing. C.M., S.C., M.S.(Marco Segarich), A.A., F.F., M.A., A.T., P.G.G., M.S. (Maurizio Spoto) and A.P.; visualization. C.M. and A.P.; supervision. A.P.; project administration. S.C. and M.S. (Maurizio Spoto); funding acquisition. A.P. All authors have read and agreed to the published version of the manuscript.

**Funding:** This research received no external funding.

**Institutional Review Board Statement:** Not applicable.

**Data Availability Statement:** Not applicable.

**Conflicts of Interest:** The authors declare no conflict of interest.

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
