# Peer review of "Haplosporidium pinnae Detection from the Faeces of Pinna nobilis: A Quick and Noninvasive Tool to Monitor the Presence of Pathogen in Early-Stage or during Fan Mussel Mass Mortalities"

_diversity, doi:10.3390/d15040477_

Round 1
Reviewer 1 Report
This article deals with DNA detection of pathogen, protozoa by PCR from faeces of endangered mussel Pinna nobilis. This method is not invasive. So, for assessment of the presence or absence of pathogen in the endangered species, this procedure is very useful. However, I notice the article at present form lack information of materials. I would like to point out the followings;
In materials and methods, how the authors did collect the faces and pseudofaeces? For example, by pipets or sediment trap, or membrane filter? Did the authors use centrifuge? Please describe the process in detail. Can authors show the amount of the materials? How did they recognize the faeces and pseudofaeces? I know it, but please suppose general readers of this journal.
The authors showed the details of samples in Table 1. These samples were derived from one mussel? For example, F2, F3 was collected at the same site and same day. F2 and F3 were different individuals? Please make it clear. I would like to ask the size of the mussels.
The published article 21 in the reference already showed the detection of H. pinnae for faces of P. nobilis . The authors should strengthen the novelty of the present study. What are new findings or new procedure? If they would work on the same samples (F8) by primers (21 in reference) and in this study, the novelty became more significant.
Author Response
This article deals with DNA detection of pathogen, protozoa by PCR from faeces of endangered mussel Pinna nobilis. This method is not invasive. So, for assessment of the presence or absence of pathogen in the endangered species, this procedure is very useful. However, I notice the article at present form lack information of materials. I would like to point out the followings;
Q.In materials and methods, how the authors did collect the faces and pseudofaeces? For example, by pipets or sediment trap, or membrane filter? Did the authors use centrifuge? Please describe the process in detail. Can authors show the amount of the materials? How did they recognize the faeces and pseudofaeces? I know it, but please suppose general readers of this journal.
A.This is a helpful suggestion. We have detailed the procedure followed in the M&M, section 2.1 (lines 84-94). Regarding the amount of material, the DNA extraction kit required a range of 10-250 mg as starting amount of material and the collected faeces were all above this amount.
Q.The authors showed the details of samples in Table 1. These samples were derived from one mussel? For example, F2, F3 was collected at the same site and same day. F2 and F3 were different individuals? Please make it clear. I would like to ask the size of the mussels.
A.Yes, every sample comes from a different specimen. We have clarified it in the M&M. section 2.1 (lines 93-94).
Q.The published article 21 in the reference already showed the detection of H. pinnae for faces of P. nobilis. The authors should strengthen the novelty of the present study. What are new findings or new procedure? If they would work on the same samples (F8) by primers (21 in reference) and in this study, the novelty became more significant.
A.We thank the reviewer for this idea. We pointed out in the text that the novelty of our system is that our assay is suitable for eDNA analysis because of the length of the amplicon (our 102 bp versus 331 bp) (lines 241-244). This opens the possibility of applying this assay to partially degraded material and environmental matrices in general. We address this aspect in more detail in the Discussion. Although we didn't perform a second amplification with the assay from [21] because we don't have these primers in our laboratory, we verified the suitability of our assay for amplification of H. pinnae by using positive samples previously amplified with the primers published in [19] (the 1-kb system). The erratum in this article [19] stems from our warning to the authors that the published sequences of primers reported to amplify a 165 bp amplicon actually amplified a 1 kb product.

Reviewer 2 Report
A well conducted and presented investigation. Therefore, I have only a few minor comments:
Lines 38-43: Please add some reference to the description of the biological traits of P. nobilis.
Line 52: Please correct to „has been“
Line 82: Is there a reference missing?
Paragraph 2.1: Please describe the method how samples were collected in the field.
Line 87: Please replace the full stop at the end of the first bracket by a comma and convert the capital I in Sistiana to a small one.
Line 133: PLease correct to follows.
Lines 223-225: Please choose one writing of pseudofaeces throughout the manuscript.
Author Response
A well conducted and presented investigation. Therefore, I have only a few minor comments:
Q.Lines 38-43: Please add some reference to the description of the biological traits of P. nobilis.
A.References have been added, thank you for this suggestion.
Q Line 52: Please correct to „has been“
A.Thank you, we have corrected it.
Q.Line 82: Is there a reference missing?
A. My apologise, I forgot to insert a conference proceeding reporting the results of the first H. pinnae detection in the Gulf of Trieste back in 2019. I have now inserted it.
Q.Paragraph 2.1: Please describe the method how samples were collected in the field.
A.We have detailed the procedure followed in the M&M, section 2.1 (lines 84-94).
Q.Line 87: Please replace the full stop at the end of the first bracket by a comma and convert the capital I in Sistiana to a small one.
A.Thank you, we have corrected it.
Q.Line 133: PLease correct to follows.
A.Thank you, we have corrected it.
Q.Lines 223-225: Please choose one writing of pseudofaeces throughout the manuscript.
A.It is true, we have now selected one writing throughout the manuscript.

Round 2
Reviewer 1 Report
The authors should edit the followings.
L86: Three different sites were sampled in the Gulf of Trieste: Miramare (Marine Protected 86 Area), Sistiana and Panzano (see Figure 1). → At three different sites in the Gulf of ,,,,,, the samples were obtained or collected.
L88: shell size? Should be shell length , or shell height in the Bivalvia.
L93: delivered to the Genetics and Zoology Laboratory (at ambient temp)?
L99: syringe with needle or without needle?
Author Response
REVIEWER 1 (ROUND 2)
The authors should edit the followings.
L86: Three different sites were sampled in the Gulf of Trieste: Miramare (Marine Protected 86 Area), Sistiana and Panzano (see Figure 1). → At three different sites in the Gulf of ,,,,,, the samples were obtained or collected.
- Thank you, the sentence has been revised.
L88: shell size? Should be shell length , or shell height in the Bivalvia.
- Thank you, the sentence has been corrected.
L93: delivered to the Genetics and Zoology Laboratory (at ambient temp)?
- Thank you, the sentence has been revised, in fact samples were not sent at ambient temperature, but refrigerated.
L99: syringe with needle or without needle?
- Whitout needle. The syringes mentioned are sold without needle, for this reason this detail was not added at the beginning. We have now added this detail. Thank you.
